# Perceived Bullying in Physical Education Classes, School Burnout, and Satisfaction: A Contribution to Understanding Children’s School Well-Being

**DOI:** 10.3390/healthcare13111285

**Published:** 2025-05-29

**Authors:** Sinan Uğraş, Ahmet Enes Sağın, Mehmet Akif Yücekaya, Cenk Temel, Barış Mergan, Nuno Couto, Pedro Duarte-Mendes

**Affiliations:** 1Faculty of Sport Sciences, Çanakkale Onsekiz Mart University, 17020 Çanakkale, Turkey; sinanugras@comu.edu.tr; 2Faculty of Sport Sciences, Bartın University, 74100 Bartın, Turkey; enessagin@bartin.edu.tr; 3Faculty of Sport Sciences, Dicle University, 21010 Diyarbakır, Turkey; mehmet.yucekaya@dicle.edu.tr; 4Faculty of Sport Sciences, Akdeniz University, 07000 Antalya, Turkey; cenktemel@akdeniz.edu.tr; 5Faculty of Sports Sciences, Tokat Gaziosmanpaşa University, 60030 Tokat, Turkey; baris.mergan@gop.edu.tr; 6Sport Sciences School of Rio Maior, Santarém Polytechnic University (ESDRM-IPSantarém), 2040-413 Rio Maior, Portugal; ncouto@esdrm.ipsantarem.pt; 7Research Center in Sport Sciences, Health Sciences and Human Development (CIDESD), 5001-801 Vila Real, Portugal; 8Department of Sports and Well-Being, Polytechnic Institute of Castelo Branco, 6000-266 Castelo Branco, Portugal; 9Sport Physical Activity and Health Research & Innovation Center, Sprint, 2040-413 Santarém, Portugal

**Keywords:** physical education, peer bullying, school burnout, school satisfaction, well-being

## Abstract

**Aim:** This study examines the effects of peer bullying that middle school students experience in physical education classes on school burnout and school satisfaction to understand children’s well-being in this important stage of their lives. **Method:** The study was conducted with 829 students from 5th, 6th, 7th, and 8th grades in Türkiye of both genders (403 male, 426 female), with an age mean of 11.7 ± 1.16 years old. Data were collected using the Physical Education Class Perceived Bullying Scale, developed by the researchers in the present study, along with the School Burnout Scale and the School Satisfaction Scale for Children. Structural equation modeling (SEM) was employed to analyze the data and examine the direct and indirect relationships between students’ perception of bullying, school burnout and satisfaction. **Results:** The findings indicate that perceived bullying in physical education classes positively and significantly predicts school burnout (β = 0.388, *p* < 0.001), while it negatively and significantly affects school satisfaction (β = −0.122, *p* = 0.006). Moreover, significant positive relationships were found between perceived bullying and school burnout, and significant negative relationships were found between perceived bullying and school satisfaction. **Conclusions:** This study reveals that peer bullying in physical education classes significantly affects students’ school burnout and satisfaction. These findings highlight the need for developing effective strategies to prevent bullying in educational settings and promote children’s healthcare and well-being.

## 1. Introduction

Peer bullying is an increasing social problem among young people with profound consequences in educational settings. Bullying is identified as physical, verbal, or psychological harm inflicted by individuals or groups (Olweus [1]), and it comprises behaviors such as physical assault, ridicule, social exclusion, and, more recently, cyberbullying [2,3]. The global reports indicate that one in three students aged 13–15 experience bullying [4], with physical bullying being the most common [5]. These trends are also observed in Türkiye, where more than 30% of students report being victims or perpetrators of bullying [6]. Students exposed to peer bullying often experience decreased self-esteem, social withdrawal, and psychological problems such as anxiety, depression, and even post-traumatic stress [7,8,9,10,11,12,13,14,15]. These effects can persist into later stages of education and significantly undermine victims’ academic performance, social adjustment, and overall well-being.

While sport-based programs have been proposed as a potential preventive strategy—emphasizing awareness, inclusion, and social skill development [16,17]—physical education and sports (PES) classes can paradoxically become a setting where bullying is more frequent than in other school contexts. Competitive dynamics, insufficient supervision, and skill disparities often create conditions that foster bullying behaviors [18,19,20]. These experiences can negatively impact students’ physical and mental health and school satisfaction, engagement, and overall learning experience [21,22]. Peer bullying in PES classes has a different structure than bullying in other classes. Factors inherent in PES classes, such as physical performance, body image, and the visibility of individual talents, increase social comparison and competition among students. This context paves the way for bullying incidents to occur more frequently and directly. Students who exhibit low skills, especially in activities requiring physical performance, are more exposed to behaviors such as ridicule, exclusion, or humiliation by other students [23,24]. For example, Bejerot et al. [25] reported that 48.6% of adults with below-average motor skills were bullied at school and pointed out that low physical abilities increase the risk of victimization. Recent findings indicate that poor fine motor skills at age 3 were significantly associated with peer victimization at age 5. Moreover, while both fine and gross motor difficulties at age 5 showed concurrent associations with peer victimization, gross motor impairments had a stronger impact. Notably, only fine motor difficulties at age 5 were significantly linked to peer victimization reported at age 8 [26]. A recent study by Bejerot, Ståtenhag, and Glans [27] found that adults diagnosed with ADHD who recalled having below-average motor skills—reflected in poor performance in PE classes—were significantly more likely to report being victims of bullying during childhood. This association was especially strong during early school years and continued through adolescence. However, no significant relationship was found between poor motor skills and perpetrating bullying. These findings underscore the role of physical competence, particularly in PE settings, as a risk factor for peer victimization.

Students who face bullying in PES classes may feel reluctant to attend classes and feel unsafe in the school environment. Such experiences may reduce students’ commitment to school and decrease their school satisfaction. Research shows that burnout symptoms may be observed more in students who tend to avoid physical activities [28,29]. School burnout refers to students’ emotional exhaustion, cynicism toward academic tasks, and a reduced sense of accomplishment in school settings [30]. On the other hand, school satisfaction is defined as students’ overall cognitive and emotional appraisal of their school experience, including their sense of belonging and perceived support [31]. Previous research has consistently shown that exposure to peer bullying is associated with increased levels of school burnout and reduced school satisfaction among students. Studies have found that victims of bullying often report higher emotional exhaustion, lower motivation, and weaker feelings of belonging at school [32].

To deepen the theoretical foundation of this study, it is valuable to consider frameworks such as the Social–Ecological Model of Bullying and Self-Determination Theory. The Social–Ecological Model [33] emphasizes the interplay between individual, relational, and contextual factors—such as peer norms, classroom climate, and teacher oversight—in shaping bullying behavior. Meanwhile, Self-Determination Theory [34] highlights how unmet needs for autonomy, competence, and relatedness can lead to maladaptive behaviors, including peer aggression or withdrawal. Additionally, conceptualizing bullying in PE classes alongside related behaviors such as hate speech or exclusion in digital contexts may broaden the understanding of student aggression. For instance, a recent intervention by Sportelli et al. [35] demonstrated that digital intergroup contact and empathy-based activities can foster adolescents’ willingness to engage in counter-speech, offering insights for peer bullying interventions both online and in schools. It is thought that bullying occurring in PES classes may cause the students to feel more alienated from the school environment by damaging their perceptions of physical confidence. It may lead students, especially those who are victims of bullying, to be distant from physical activities and social interactions [36]. The difficulties of conducting PES classes in large physical areas and distributing teacher supervision equally throughout the area can make it challenging to detect bullying cases. This situation becomes more evident, especially in team games and competitive activities, and students who perform poorly due to individual skill differences are at risk of being targets. Such experiences may lead to adverse outcomes in students, such as feelings of exclusion and disconnection from their social environments [19,37]. The peer bullying that students face in PES classes may have different dynamics from other school environments. Intensely competitive physical activities and differences in skill levels among the students may make the risk of bullying more apparent in these classes.

The assessment of peer bullying has been made through various scales that generally focus on general types of bullying [38,39] and do not adequately reflect the competition, social comparison, and physical performance-centered nature of physical education and sport (PES) classes. The specific dynamics of the sports context necessitate the development of a bullying scale specific to PES. Such a scale would allow for measuring the effects of bullying on students’ physical self-esteem, body image, perception of social acceptance, as well as school satisfaction and burnout. Therefore, considering this limitation and the negative implications of peer bullying in school burnout and school satisfaction of this population, we have developed the “Perceived Peer Bullying Scale in Physical Education Class”, which allows for the assessment of perceived bullying in the physical education class. Using this tool, this study aims to examine the effects of peer bullying that middle school students perceive in physical education classes on school burnout and school satisfaction.

This study is expected to provide valuable insights into the dynamics of bullying in physical education and sports classes and contribute to developing effective intervention strategies to address and prevent bullying in these settings.

## 2. Methods

### 2.1. Participants

Data were collected from two independent samples. The first sample, for the development of the “Physical Education Classes Perceived Bullying Scale”, comprised a total of 225 (age: 11.5 ± 1.12) secondary school students, 59.1% of whom were female (*n* = 133; age: 11.6 ± 1.14) and 40.9% of whom were male (*n* = 92; age: 11.4 ± 1.09). Of the participants, 24.9% were 5th-grade students (*n* = 56), 27.1% were 6th-grade students (*n* = 61), 23.1% were 7th-grade students (*n* = 52), and 24.9% were 8th-grade students (*n* = 56). This sample was considered for preliminary translation and validation of the Physical Education Class Perceived Bullying Scale. To obtain reliable results in exploratory factor analysis, it is indicated that it is necessary to work with at least 200 to 300 participants [40] or at least five times the total number of scale items [41]. In this context, it can be stated that the first sample group was sufficient for scale development.

After the scale was developed, the data of the actual research group were collected. The required sample size was estimated using Daniel Sopper’s online calculator [42] based on the following criteria: an expected effect size of 0.1, a statistical power of 0.80, three latent variables, and 25 observed variables. As a result, the calculator suggested a minimum of 119 participants to detect the effect and 823 for the structural model. In line with these results, the present study employed a sample size of 823, fulfilling the recommended threshold. Data from the second sample comprised a total of 829 (age: 11.7 ± 1.16) students in Türkiye, 48.6% of whom were male students (*n* = 403; age: 11.6 ± 1.09) and 51.4% were female students (*n* = 426; age: 11.6 ± 1.15). Of the participants, 21% were 5th-graders (*n* = 174; age: 10.1 ± 0.254), 25.1% were 6th-graders (*n* = 208; age: 11.0 ± 0.241), 27.6% were 7th-graders (*n* = 229; age: 12.0 ± 0.175), and 26.3% were 8th-graders (*n* = 218; age: 13.0 ± 0.280). This sample was considered for the confirmatory factor analysis of the Physical Education Class Perceived Bullying Scale and to investigate the Relationship Between Physical Education Class Perceived Bullying, School Burnout and School Satisfaction. The data were collected from public schools in the provincial centers of five different regions of Türkiye, representing average socio-demographic characteristics. The study protocol was approved by the Ethical Committee report from Çanakkale Onsekiz Mart University, dated 30 May 2024, with the number 08/30, to establish the ethical principles for data collection.

### 2.2. Data Collection

Data collection was carried out in accordance with the principles stated in the Declaration of Helsinki [43] and approved by Ethics Committee report number 08/30 from Çanakkale Onsekiz Mart University, dated 30 May 2024. The completion time for all data collection procedures was approximately 10 min. Permission was obtained from the school principals before data collection began. A signed informed consent form was given to the parents of the participating students before data collection. The explanation letter was read before data collection. Then, one of the leading researchers collected the data anonymously from the participants during school hours in the Physical Education classes of secondary school students.

### 2.3. Instruments

#### 2.3.1. Physical Education Class Perceived Bullying Scale (PECPB)

The researchers developed the scale for this study to determine the bullying levels that secondary school students perceived in Physical Education classes. The literature was first examined in the scale development process, and then an item pool was created. Then, the 31 items were sent to experts who had previously conducted studies on peer bullying. The opinions of three experts were obtained using the Lashwe [44] technique. The Likert scale is a 5-point scale with the following options: ‘Never’, ‘Rarely’, ‘Occasionally’, ‘Very often’ and ‘Always’. Higher average scores indicate greater perceived peer bullying in physical education classes. An example is the following: “My classmates make fun of my performance during physical education classes”. After removing the items that were not suitable according to expert opinion, the remaining 23 items were applied to the group to be subjected to Exploratory Factor Analysis (EFA), with the first sample group (*n* = 225) and the factorial structure of the questionnaire adaptation examined using the principal components extraction method with Promax Rotation [45]. Bartlett’s Test of Sphericity results were detected as (χ^2^ (45) = 1613, (*p* < 0.001)), and the KMO value was detected as 0.910. These data showed that it was suitable for factor analysis and that the sampling adequacy was high. Thirteen items with item factor loadings below 0.40 were removed from the data set [46]. The remaining ten items were found to vary between 0.520 and 0.920, which, according to Cid et al. [47], are considered significant since the value is equal to or greater than 0.5 (FL ≥ 0.50). As a result of the exploratory factor analysis, a single-factor structure was determined using the minimum residuals method and varimax rotation. It was identified that the eigenvalue of the first factor was 5.6742, and the others were below 1, and this single-factor structure explained 56.7% of the total variance. It is stated in the literature that the explained variance ratio should be between 40% and 60% [48]. Internal reliability was assessed using the Cronbach’s alpha coefficient. Cronbach’s alpha for the scale was 0.917, indicating excellent internal consistency [49]. After the exploratory factor analysis results, confirmatory factor analysis was performed with the second sample (*n* = 829). The maximum likelihood estimation method was employed. Since the assessment of multivariate kurtosis coefficient indicator was greater than 5, Bollen–Stin (B-S) Bootstrapping (2000 re-samples) was used to test the structure of the data obtained [50,51]. According to the CFA analysis results (Figure 1), it was determined that the items varied between standardized β = 0.567 and β = 0.814, X^2^ = 284/df = 35, *p* < 0.001, pB-S = <0.001, CFI = 0.938, TLI = 0.921, GFI = 0.931, SRMR = 0.042 and RMSEA = 0.093, values which were within acceptable limits [51]. Average Variance Extracted (AVE) was calculated to determine how much variance the items of the construct could explain, and Construct Reliability (CR) was calculated to evaluate the internal consistency and reliability of the items. While the AVE value was calculated as 0.491, the CR value was 0.906. If the AVE value is below 0.50, construct validity can be accepted if the CR value is above 0.70 [52]. The Cronbach’s Alpha value of the measurement tool was found to be 0.905. These results show that the measurement tool is valid and reliable. Statistical analyses were performed using IBM SPSS and IBM AMOS (SPSS Inc., Chicago, IL, USA, version 28.0).

#### 2.3.2. School Burnout Scale (SB)

The scale developed by Salmela-Aro et al. [30] to determine the school burnout levels of students was adapted to Turkish culture by Secer et al. [53]. The scale consists of nine items, including four items for “emotional exhaustion”, three items for “depersonalization”, and two items for “low expectation of success”. Items are rated on a 5-point Likert scale ranging from 1 (Strongly disagree) to 5 (Strongly agree), with higher scores indicating a greater level of school burnout. A sample item from the scale is: “I feel overwhelmed by schoolwork”. In line with the hypotheses formed in this study, a secondary-level CFA analysis was conducted to test its unidimensional structure. According to the CFA results, the item factor loadings ranged between β = 0.563 and β = 0.799, X^2^ = 122/df = 24, *p* ≤ 0.1, CFI = 0.973, TLI = 0.959, IFI = 0.973, GFI = 0.985, SRMR = 0.026, and RMSEA was found as 0.070. Cronbach’s Alpha value was calculated as 0.899. According to these results, the School Burnout Scale has valid and reliable values.

#### 2.3.3. Overall School Satisfaction Scale for Children (SS)

The scale for measuring the comprehensive school satisfaction of primary and secondary school children was adapted into Turkish by Telef [54]. The scale uses a 5-point Likert format, with six items and one dimension. Higher average scores indicate overall school satisfaction. As a result of the CFA analysis conducted to test the structure, modifications were made because the X^2^/df and RMSEA values were not within acceptable limits. As a result of the covariances between the 2nd and 4th items and 4th and 6th items, X^2^ = 51.4/df = 7, *p* < 0.001, CFI = 0.986, TLI = 0.969, IFI = 0.986, GFI = 0.997, SRMR = 0.018 and RMSEA = 0.087 values were determined. Cronbach’s Alpha value of the scale was determined as 0.911. With these values, it can be stated that the results are acceptable.

### 2.4. Statistical Analyses

As a first step, the study calculated descriptive statistics such as means and standard deviations. Furthermore, distribution characteristics (skewness and kurtosis) and bivariate correlations among the variables were examined. In the normality test, data are considered normal if the distribution of values is between −2.0 to 2.0 for skewness and −7.0 to 7.0 for kurtosis [49]. Pearson correlation analysis was performed to determine the relationship between the variables. Confirmatory Factor Analysis was conducted to test the model. The maximum likelihood estimation method was employed since the value of the multivariate kurtosis coefficient indicator assessment was greater than 5 in the Bollen-Stine Bootstrapping (2000 re-samples) [50,51]. The fitness of both the measurement and structural models was assessed using commonly accepted incremental fit indices, including the Comparative Fit Index (CFI) and Tucker–Lewis Index (TLI), as well as absolute fit indices such as the Standardized Root Mean Residual (SRMR) and the Root Mean Square Error of Approximation (RMSEA), accompanied by its confidence interval. In evaluating model adequacy, the threshold values proposed by the prior literature [49,55] were followed: CFI and TLI values of 0.90 or higher and RMSEA and SRMR values of 0.08 or lower were considered indicative of good fit. Although the chi-square (χ^2^) to degrees of freedom (df) ratio is presented for completeness, it was not used as a primary evaluation criterion due to its known sensitivity to model complexity [49]. Convergent validity was examined through the Average Variance Extracted (AVE), where values equal to or exceeding 0.50 were interpreted as satisfactory [49].

## 3. Results

Table 1 shows the descriptive statistics; the mean of physical education class perceived bullying (PECPB) was M = 1.78 (SD = 0.91), with a skewness value of 1.55 and a kurtosis value of 2.09. School Satisfaction (SS) shows values of mean M = 4.00 (SD = 1.08), skewness −1.09, and kurtosis value 0.55. The School Burnout (SB) variable has an M = 2.22 (SD = 1.05) mean value with a skewness value of 0.82 and a kurtosis value of −0.06. These statistical values indicate that the distributions of the variables are within the limits of normality. A negative and significant relationship was found between physical education class perceived bullying (PECPB) and school satisfaction (SS) (r = −0.099, *p* < 0.001). This indicates that as perceived bullying increases, school satisfaction decreases. A positive and significant relationship was found between physical education class perceived bullying (PECPB) and school burnout (SB) (r = 0.359, *p* < 0.001), indicating that as the perception of bullying increases, school burnout also increases. In addition, there is a strong and negative relationship between school satisfaction and school burnout (r = −0.446, *p* < 0.001); according to this finding, as school satisfaction increases, school burnout decreases.

The research model was tested with a measurement model. The measurement model plays a critical role in accurately representing abstract concepts with concrete indicators [56] and assessing the reliability and validity of the scales used [57]. The measurement model was preferred because it solves complex relationships, minimizes measurement errors, and thus obtains more reliable results [52]. Figure 2 presents the ten indicator items of the physical education class perceived bullying scale that were analyzed. Factor loadings ranged from β = 0.566 to β = 0.81, and all factor loadings were significant (*p* < 0.001). The AVE (Average Variance Extracted) value for this construct was calculated as 0.501, and the construct validity was sufficient. In addition, Cronbach’s Alpha value of the physical education class perceived bullying dimension was found to be 0.907, indicating that the scale has a high internal consistency. The six indicator items of the school satisfaction scale were analyzed. Factor loadings ranged from β = 0.760 to β = 0.848, and all factor loadings were significant (*p* < 0.001). The AVE value for this dimension was calculated as 0.633, indicating that a considerable portion of the variance can be explained and that this construct is valid. Moreover, Cronbach’s Alpha value of the school satisfaction dimension was calculated as 0.911, indicating a high reliability level. The nine indicator items of the school burnout scale were analyzed. Factor loadings ranged from β = 0.548 to β = 0.787, and all loadings were significant (*p* < 0.001). The AVE value of the school burnout dimension was found to be 0.507, and the construct validity was found to be acceptable. The Cronbach’s Alpha value of this dimension was calculated as 0.900, indicating that the scale has high internal consistency. The values of the measurement model X^2^ = 1138/df = 273, *p* ≤ 0.001, p B-S = 0.005, CFI = 0.923, TLI = 0.915, GFI = 0.901 and RMSEA = 0.055 were found to be within acceptable limits [50]. These results show that the measurement model is reliable.

Regarding the effects of physical education class perceived bullying (PECPB) on school burnout (SB) and school satisfaction (SS), the results show that the impact of physical education class perceived bullying on school burnout was positive and significant (β = 0.388, SE = 0.0472, z = 9.36, *p* < 0.001). This finding demonstrates that school burnout increases as the perception of bullying increases. In contrast, the effect of physical education class-perceived bullying on school satisfaction was negative and significant (β = −0.122, SE = 0.0451, z = 0–2.73, *p* = 0.006). This result shows that as the bullying students perceive in physical education classes increases, their school satisfaction decreases.

## 4. Discussion

This study examines the effects of perceived peer bullying in physical education and sports (PES) classes on middle school students’ school burnout and satisfaction. The findings of the study show that bullying experiences in PES classes weaken students’ commitment to school and increase their school burnout levels. While these educational outcomes are distinct from broader psychological and social health measures, the findings align with the existing literature emphasizing the negative impact of peer bullying on various aspects of student well-being [30,58,59,60]. This study contributes to the literature by offering a class-specific perspective. It suggests that bullying may have unique manifestations and consequences in physical education and sports (PES) classes due to its competitive, performance-based, and socially visible nature. In such settings, where physical competence is prominent and the need for social approval is heightened, bullying appears to affect students’ experiences of school satisfaction and burnout particularly. These findings provide evidence that perceived bullying in PE settings significantly contributes to increased school burnout and decreased school satisfaction by undermining students’ sense of belonging, self-worth, and perceived competence [61,62,63]. However, it is important to note that this study did not include data from other subject areas; therefore, conclusions about the distinctiveness of PE-related bullying outcomes should be interpreted cautiously. Future research could benefit from cross-subject comparisons to better understand whether the dynamics and effects of bullying in PE classes are indeed distinct from those observed in other educational settings.

The study results are consistent with previous research emphasizing the adverse effects of peer bullying on school satisfaction. In their studies, Arslan et al. [64] and Smokowski et al. [65], who indicated that bullying is a damaging factor in students’ school experiences, show that bullying can reduce school satisfaction and student engagement. The increased risk of bullying in environments with intense social comparison and competition, such as PES classes, suggests that students’ school satisfaction may be particularly low in these classes. Recent research has also highlighted that peer victimization in school settings can reduce students’ life satisfaction and academic self-efficacy. For instance, a large-scale study among middle school students in Switzerland showed that supportive school environments and a strong sense of peer belonging were associated with reduced victimization and higher life satisfaction [66]. Chen et al. [67] conducted a large-scale, cross-national study using PISA 2015 and 2018 data from five countries (China, Japan, South Korea, the US, and the UK) to investigate how peer victimization affects adolescents’ school belonging, truancy, and life satisfaction. The findings revealed that peer victimization had a significant negative direct effect on school belonging, which in turn mediated both increased truancy and decreased life satisfaction. Huang [68] found that peer victimization was negatively associated with students’ academic life satisfaction and school belonging and positively associated with schoolwork-related anxiety. These findings are consistent with the current study’s results, highlighting how peer bullying in school contexts can reduce students’ emotional engagement and overall satisfaction with school. 

On the other hand, our study revealed that bullying in PES classes had a significant effect on students’ school burnout. School burnout may occur as a result of the psychological stress and feelings of inadequacy that students experience when they are exposed to bullying [69,70]. However, the effects of bullying are not limited to psy-chological exhaustion but can also severely undermine students’ academic motivation and interest in learning [71,72]. Bullying in PES classes can further deepen the feelings of inadequacy, especially in this environment where students’ physical abilities come to the fore, and social comparison among peers is intense. Such social comparison and perception of inadequacy can undermine students’ self-esteem, setting the stage for psychological and academic burnout [73,74].

Furthermore, research suggests that students’ perceptions of their physical abilities may be negatively affected as a result of bullying by their peers, which may increase body image problems and academic burnout in the long run [75,76]. For example, earlier research by Puhl and Luedicke [76] indicated that students who experienced bullying related to their physical abilities tended to avoid physical activities and disengage from general school participation. Supporting this, Benítez-Sillero et al. [77] found that lower levels of cardiorespiratory fitness were significantly associated with victimization, whereas higher muscular strength—particularly in boys—was positively related to bullying perpetration. The competitive nature of PES classes and the fact that students who perform poorly in these classes attract more attention may lead students to experience a general feeling of social exclusion in the school environment, which may increase their burnout levels. In this context, our findings emphasize that bullying in PES classes should be addressed as an important risk factor that increases student burnout.

Our study contradicts some research emphasizing the potential of sport-based practices in reducing peer bullying. For example, it has been suggested that sports activities can increase social skills, strengthen teamwork, and thus effectively reduce bullying [23,37,78]. These and similar studies assert that sports can effectively prevent bullying behaviors by improving students’ social skills. However, the findings obtained in our study indicate that the structure of PES classes, where competition and social comparison are at the forefront, may turn these classes into a risky context in terms of bullying. Namely, while PES classes have the potential to provide socialization and solidarity, they also provide an environment where students’ physical abilities become visible and where comparison and competition are intense. This dual structure may increase the risk of exposure to bullying, especially for students with low physical abilities.

These contradictory findings suggest that, despite the potential of PES classes to develop social skills, they may create a suitable environment for bullying due to their competitive nature. Although it is stated that sports can be an effective tool in dealing with bullying when the right strategies and a safe environment are provided in the research [79,80], it is emphasized that sports activities may have different results when they turn into a competitive and skill-based structure. For instance, students who perform poorly in PES classes may encounter negative behaviors such as exclusion, being belittled, or being ridiculed by their more able peers [81]. Such experiences may reduce the students’ commitment to the school environment by undermining their academic and psychological motivation. It is vital to delve deeper into how bullying in physical education (PE) contributes to student burnout and reduced satisfaction. As demonstrated by Brito and Oliveira [82], involvement in bullying—whether as a victim, aggressor, or observer—is significantly associated with lower self-esteem, particularly among female students. This diminished sense of self-worth can directly impact students’ emotional resilience and increase their vulnerability to emotional exhaustion.

Furthermore, the findings of Jachyra [83] suggest that repeated experiences of explicit and symbolic humiliation in PE not only lead to withdrawal from participation but also contribute to a broader cultural disaffection toward health and physical education among adolescent boys. Together, these insights underscore how bullying undermines both the intrinsic motivation and psychological well-being of students, which are critical drivers of active and sustained participation in physical activity. Low self-esteem, in particular, may mediate the relationship between bullying and disengagement, amplifying feelings of incompetence and alienation within the PE context. To translate these findings into practical action, schools could benefit from developing clear anti-bullying policies specific to PE environments, integrating teacher training programs that promote inclusive and empathetic pedagogical approaches, and implementing student-centered interventions to build peer support networks and enhance critical health literacy. Future research should explore the efficacy of such interventions, especially those that explicitly aim to rebuild self-esteem and foster a positive identity through physical activity participation.

## 5. Limitations

This study has some limitations. First, since the study’s data are based on students’ self-reports, the sincerity and accuracy of the responses may be limited. Furthermore, since the study was limited to middle school students in Türkiye, the generalizability of the findings may be limited to different cultural contexts. In addition, the study did not include background variables such as students’ physical fitness, body image perception, or motor skill levels. These characteristics may be associated with bullying experiences in PE settings, and their exclusion limits the depth of interpretation regarding individual differences. Future research should integrate such variables to provide a more holistic understanding of perceived bullying and student well-being in physical education contexts. These factors should be considered for the universal validity of the study’s findings. However, the comprehensive perspective the study offers to understand the dynamics of bullying specific to PES classes is an important contribution of the research. The bullying scale explicitly developed for PES classes allows for a more detailed and specific assessment of bullying cases in this context. The study also addresses the effects of bullying in PES classes on school burnout and satisfaction, suggesting that these classes should be re-evaluated in the education system.

## 6. Future Directions

In future research, it would be useful to examine bullying in PES classes in different cultural contexts. Furthermore, assessing the long-term effects of bullying in PES classes may provide a way to deal with the consequences in terms of students’ psychosocial health in a more comprehensive way. Examining the effectiveness of bullying prevention programs for PES classes through longitudinal studies can significantly contribute to ensuring that these classes provide a safe and supportive environment. Research on the role of PES classes in educational policies will allow us to better understand the effects of these classes on the general development of students. Future studies should consider incorporating behavioral indicators (e.g., facial expressions, gaze tracking, or physiological measures) to capture student engagement and stress levels more objectively and to mitigate potential social desirability bias in self-reported data [84,85].

## 7. Conclusions

Through the main results of this work, it was identified that the perceived bullying levels of secondary school students in physical education classes have a positive and significant effect on school burnout and a negative and significant effect on school satisfaction. As a result, it is understood that despite the potential of PES classes to provide social skills and encourage solidarity, these classes can create a suitable environment for bullying if not structured correctly. Our research has shown that the structure of PES classes, where competition, physical performance, and social comparisons are prominent, may create psychological and social pressure on students and trigger negative effects such as school burnout. Therefore, it is crucial for PES teachers to plan their class content inclusively, balance social comparison and competition, and use strategies that strengthen social acceptance among students in order to minimize the risk of bullying.

## Figures and Tables

**Figure 1 healthcare-13-01285-f001:**
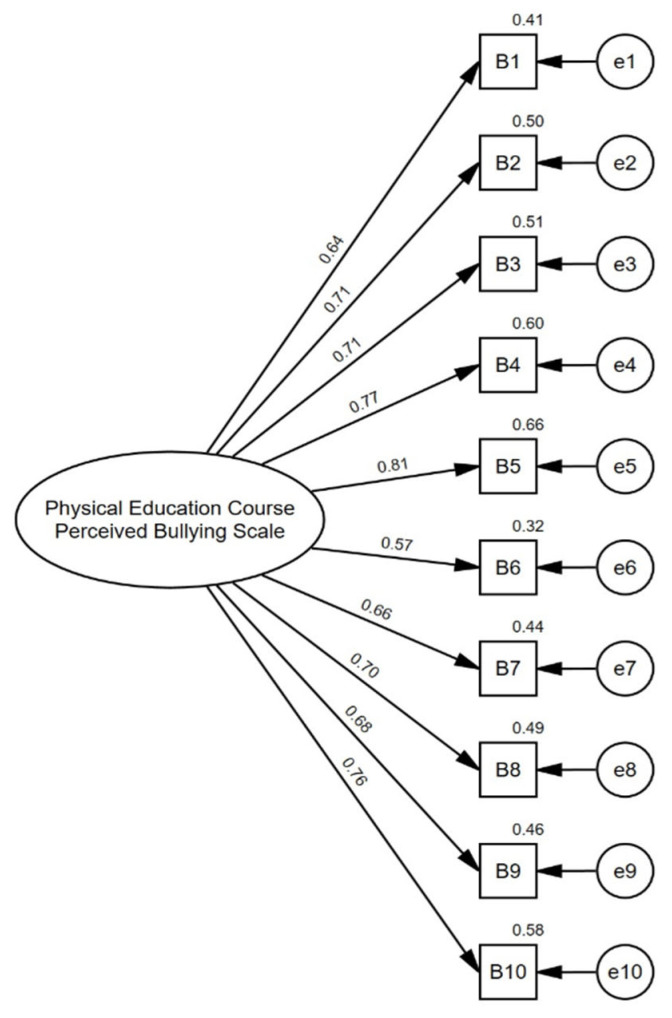
Physical Education Class Perceived Bullying Scale–standardized individual parameters.

**Figure 2 healthcare-13-01285-f002:**
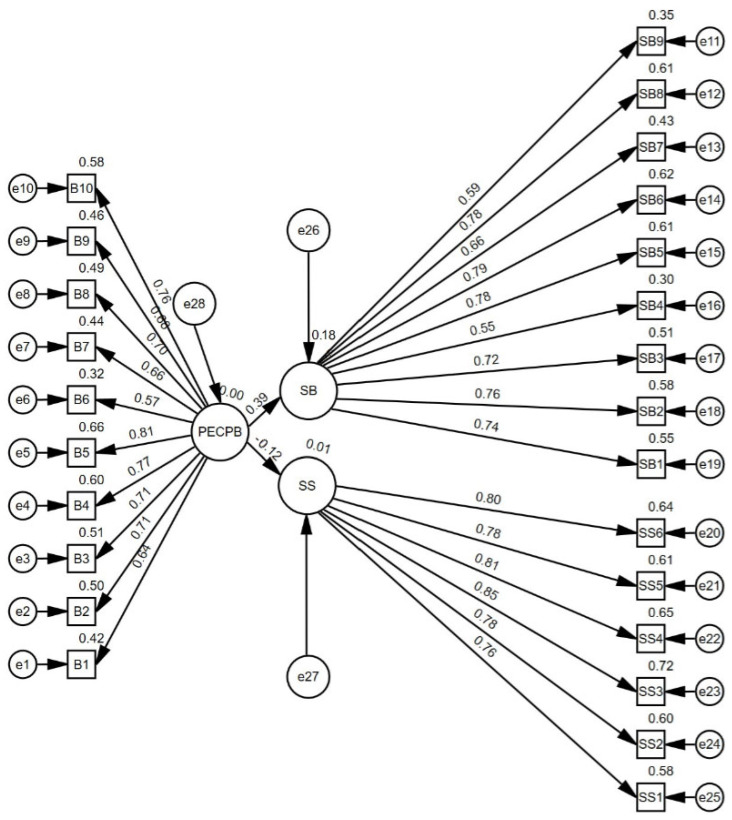
The prediction level of physical education class perceived bullying on school satisfaction and burnout—Standardized individual parameters. Notes: PECPB—Physical Education Class Perceived Bullying Scale; SS—Overall School Satisfaction Scale for Children; SB—School Burnout Scale.

**Table 1 healthcare-13-01285-t001:** Mean, standard deviation, and correlations between constructs.

Variables	PECPB	SS	SB	Mean	SD	Skewness	Kurtosis
PECPB	-			1.78	0.906	1.55	2.09
SS	−0.099 **	-		4.00	1.08	−1.09	0.548
SB	0.359 **	−0.446 **	-	2.22	1.05	0.824	−0.0648

Notes: PECPB—Physical Education Classes Perceived Bullying Scale; SS—Overall School Satisfaction Scale for Children; SB—School Burnout Scale; SD—standard deviations; **—*p* < 0.001.

## Data Availability

The datasets used and analyzed during the current study are available from the corresponding author on reasonable request.

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
