# Peer review of "Perceived Bullying in Physical Education Classes, School Burnout, and Satisfaction: A Contribution to Understanding Children’s School Well-Being"

_healthcare, 2025, doi:10.3390/healthcare13111285_

Round 1
Reviewer 1 Report
Comments and Suggestions for Authors
The study addresses a critical issue—peer bullying in physical education—and its impact on school burnout and satisfaction, contributing to research on child well-being. However, a stronger theoretical grounding would enhance the study. Integrating models such as the Social-Ecological Model of Bullying or Self-Determination Theory could provide deeper insights. Additionally, the study should compare bullying in PE with similar behaviors, such as hate speech targeting ethnic minorities, and explore potential interventions (see, for instance, Sportelli et al., 2025, on counter-speech interventions in adolescence), by taking into account that potential intervention on peer bullying can be compared with the discredit of people on digital settings.
Regarding the methodology, the use of a large and diverse sample (N=829) strengthens the reliability of the findings, and Structural Equation Modeling (SEM) allows for a sophisticated analysis of direct and indirect effects. The development of the Physical Education Course Perceived Bullying Scale adds originality and relevance. However, the authors should provide more details on scale validation (e.g., Cronbach’s alpha for reliability) and SEM model fit indices (e.g., RMSEA, CFI, TLI) to clarify the robustness of the statistical analysis.
Moreover, future studies should incorporate behavioral measures to assess students’ engagement or stress during the task/questionnaire, including in digital settings. This approach could help identify potential social desirability biases in responses (see for instance: Carolis, B et al (2019, October). “Engaged Faces”: Measuring and Monitoring Student Engagement from Face and Gaze Behavior. In IEEE/WIC/ACM International Conference on Web Intelligence-Companion Volume (pp. 80-85), on measuring engagement through facial and gaze behavior, and Tomaka et al., 1992, on social desirability and stress responses).
In the discussion, the authors could further explore the mechanisms through which bullying in PE contributes to burnout and reduced satisfaction—for example, by examining its effects on self-esteem, motivation, or participation in physical activity. Additionally, more practical recommendations (e.g., school policies, teacher interventions) would make the findings more actionable, alongside proposals for future intervention strategies targeting students.
Author Response
Dear Reviewer
Manuscript ID:
healthcare-3540329
Manuscript title:
Perceived bullying in physical education classes, school burnout and satisfaction: a contribution to understand – Children school well-being
My colleagues and I would like to thank you for the opportunity to resubmit our manuscript to Healthcare. We found that the academic reviewers’ comments were very helpful, and we have done our best to incorporate all their suggestions and reply to the reviewers’ comments. We believe that this has made a significant contribution to the overall quality of the manuscript.
Thank you for considering our manuscript.
On behalf of all authors, yours sincerely,
Comments from Reviewer 1
The study addresses a critical issue—peer bullying in physical education—and its impact on school burnout and satisfaction, contributing to research on child well-being.
Authors' response: Thank you for the valuable suggestions for improving our paper. We have addressed each of your comments and revised the manuscript, highlighting the changes in yellow.
However, a stronger theoretical grounding would enhance the study. Integrating models such as the Social-Ecological Model of Bullying or Self-Determination Theory could provide deeper insights. Additionally, the study should compare bullying in PE with similar behaviors, such as hate speech targeting ethnic minorities, and explore potential interventions (see, for instance, Sportelli et al., [35] on counter-speech interventions in adolescence), by taking into account that potential intervention on peer bullying can be compared with the discredit of people on digital settings.
Authors' response:
Thank you for this insightful and constructive suggestion. In response, we have strengthened the theoretical framework in the introduction by incorporating references to the Social-Ecological Model of Bullying and Self-Determination Theory to provide a more comprehensive understanding of the mechanisms underlying peer bullying. Additionally, we have added a comparative perspective by discussing bullying alongside related behaviors such as hate speech, and we referenced the recent study by Sportelli et al. [35], which explores counter-speech interventions. We believe these additions enhance both the depth and relevance of the theoretical grounding of the study.
“To deepen the theoretical foundation of this study, it is valuable to consider frameworks such as the Social-Ecological Model of Bullying and Self-Determination Theory. The Social-Ecological Model [33] emphasizes the inter-play between individual, relational, and contextual factors such as peer norms, classroom climate, and teacher oversight in shaping bullying behavior. Meanwhile, Self-Determination Theory [34] highlights how unmet needs for autonomy, competence, and relatedness can lead to maladaptive behaviors, including peer aggression or withdrawal. Additionally, conceptualizing bullying in PE classes alongside related behaviors such as hate speech or exclusion in digital contexts may broaden the understanding of student aggression. For instance, a recent intervention by Sportelli et al. [35] demonstrated that digital intergroup contact and empathy-based activities can foster adolescents’ willingness to engage in counter-speech, offering insights for peer bullying interventions both online and in schools.”
Regarding the methodology, the use of a large and diverse sample (n=829) strengthens the reliability of the findings, and Structural Equation Modeling (SEM) allows for a sophisticated analysis of direct and indirect effects. The development of the Physical Education Course Perceived Bullying Scale adds originality and relevance. However, the authors should provide more details on scale validation (e.g., Cronbach’s alpha for reliability) and SEM model fit indices (e.g., RMSEA, CFI, TLI) to clarify the robustness of the statistical analysis.
Authors' response: We are thankful for the reviewer’s comment. The development of the Physical Education Course Perceived Bullying Scale scale validation results are in lines 179-188
“According to the CFA analysis results, it was determined that the items varied between standardized β=0.567 and β=0.814, X² =284/df = 35, p<.001, pB-S=<0.001, CFI=0.938, TLI=0.921, GFI=0.931, SRMR=0.042 and RMSEA=0.093” “Average Variance Extracted (AVE) was calculated to determine how much variance the items of the construct could explain, and Construct Reliability (CR) was calculated to evaluate the internal consistency and reliability of the items. While the AVE value was calculated as 0.491, the CR value was 0.906. It is stated that if the AVE value is below 0.50, construct validity can be accepted if the CR value is above 0.70 [52]. The Cronbach's Alpha value of the measurement tool was found to be 0.905.”
Moreover, future studies should incorporate behavioral measures to assess students’ engagement or stress during the task/questionnaire, including in digital settings. This approach could help identify potential social desirability biases in responses (see for instance: Carolis, B et al. [82]. “Engaged Faces”: Measuring and Monitoring Student Engagement from Face and Gaze Behavior. In IEEE/WIC/ACM International Conference on Web Intelligence-Companion Volume (pp. 80-85), on measuring engagement through facial and gaze behavior, and Tomaka et al. [81], on social desirability and stress responses).
Authors' response:
Thank you for this insightful suggestion. We fully agree that incorporating behavioral measures such as facial expressions, gaze behavior, or physiological responses could significantly enhance the assessment of student engagement and help address potential social desirability bias. As the current study was primarily based on self-report questionnaires, integrating behavioral data was beyond its methodological scope. However, we acknowledge the value of such an approach, especially in digital learning environments, and have incorporated this point into the “Limitations and Future Research” section of the manuscript.
“Future studies should consider incorporating behavioral indicators (e.g., facial expressions, gaze tracking, or physiological measures) to capture student engagement and stress levels more objectively, and to mitigate potential social desirability bias in self-reported data [81,82].
In the discussion, the authors could further explore the mechanisms through which bullying in PE contributes to burnout and reduced satisfaction, for example, by examining its effects on self-esteem, motivation, or participation in physical activity. Additionally, more practical recommendations (e.g., school policies, teacher interventions) would make the findings more actionable, alongside proposals for future intervention strategies targeting students.
Authors' response:
Thank you for your valuable feedback. In response to your suggestion, we have expanded the discussion section to more thoroughly explore the mechanisms by which bullying in physical education (PE) classes contributes to student burnout and reduced satisfaction. Drawing on relevant literature [79,80], we now discuss how repeated exposure to bullying negatively impacts students’ self-esteem, intrinsic motivation, and willingness to participate in physical activity factors that are key to student well-being and sustained engagement in PE. Additionally, to enhance the practical relevance of our findings, we have included specific and actionable recommendations. These include the implementation of anti-bullying policies tailored to PE settings, professional development for teachers on inclusive pedagogical strategies, and student centered interventions that promote peer support and critical health literacy. We also propose directions for future research to evaluate the effectiveness of such interventions in restoring self-esteem and improving participation outcomes. We sincerely appreciate your comments, which helped us strengthen both the theoretical depth and applied significance of our study.
“. It is important to delve deeper into the mechanisms by which bullying in physical education (PE) contributes to student burnout and reduced satisfaction. As demonstrated by Brito and Oliveira [79], involvement in bullying whether as a victim, aggressor, or observer is significantly associated with lower self-esteem, particularly among female students. This diminished sense of self-worth can directly impact students’ emotional resilience and increase their vulnerability to emotional exhaustion. Furthermore, the findings of Jachyra [80] suggest that repeated experiences of explicit and symbolic humiliation in PE not only lead to withdrawal from participation but also contribute to a broader cultural disaffection toward health and physical education among adolescent boys. Together, these insights underscore how bullying undermines both the intrinsic motivation and psychological well-being of students, which are critical drivers of active and sustained participation in physical activity. Low self-esteem, in particular, may mediate the relationship between bullying and disengagement, amplifying feelings of incompetence and alienation within the PE context. To translate these findings into practical action, schools could benefit from developing clear anti-bullying policies specific to PE environments, integrating teacher training programs that promote inclusive and empathetic pedagogical approaches, and implementing student-centered interventions aimed at building peer support networks and enhancing critical health literacy. Future research should explore the efficacy of such interventions, especially those that explicitly aim to rebuild self-esteem and foster a positive identity through physical activity participation.”
Reviewer 2 Report
Comments and Suggestions for Authors
Dear Authors,
Thank you for the opportunity to review the topic of the manuscript, “Perceived Bullying in Physical Education Classes, School Burnout and Satisfaction: A Contribution to Understand –Children School Well-Being.”, addresses an important issue regarding students' psychological experiences in school. The objective to examine the impact of peer bullying in physical education (PE) classes on school burnout and satisfaction among middle students is valuable. However, there are several concerns related to the structure, clarity, methodology, and interpretation of results that need to be addressed to enhance the rigor and focus of the manuscript.
- Manuscript Structure
- The manuscript should follow the conventional structure: Introduction, Methods, Results, Discussion, and Conclusion.
- Sections currently labeled as 3. Procedures, 4. Instruments, and 5. Statistical Analyses should be consolidated under section 2. Methods.
- Introduction
- Line 49: The citation format is incorrect. It should be written as (Olweus, 1973, 1978).
- The impact of peer bullying is repeated in the second and third paragraphs—please revise to avoid redundancy.
- The statement “The literature suggests that sport-based practices may have a potential impact on preventing peer bullying” lacks a supporting citation—please provide references.
- The claim that “Peer bullying in PES courses has a different structure than bullying in other courses” is unclear, especially since the forms of bullying mentioned (physical, verbal, psychological) are similar. Please clarify and support this claim with literature.
- Key terms such as School Burnout and School Satisfaction are not defined. A thorough literature review explaining the empirical relationships among the study variables is also missing.
- It is unclear whether the School Burnout Scale, which includes subscales such as “emotional exhaustion,” “depersonalization,” and “low expectation of success,” accurately captures the construct of school burnout in this study. Please justify the choice of scale.
- Methods
- Participants: While participant information is described for the first stage, the second stage includes the primary sample. Demographic characteristics should be moved to the Results section.
- Lines 172–173: These lines contain duplicate content and should be revised.
- Instruments:
- PECPB Scale: The development process is described, but the actual items and the types of bullying measured are not presented. It is unclear whether the scale includes dimensions such as physical, verbal, psychological, or cyberbullying. The scoring method and interpretation of scores must also be clarified.
- School Burnout Scale and School Satisfaction Scale: Please report the Cronbach’s alpha values obtained in this study. Provide one sample item from each subscale. Scoring methods and score interpretations should be clearly explained.
- Line 226: Correct the typo from “theSopper´s online” to “the Sopper´s online.”
- The content under 5.1 Preliminary Analysis should be moved to the Participants section as it explains sample size estimation.
- Results
- Lines 260–271: This section should focus solely on statistical results. Descriptions of statistical methods belong in the Statistical Analysis section.
- The results should be presented in accordance with the research objective, beginning with the participants’ demographic characteristics, followed by the status of peer bullying, school burnout, and school satisfaction. Only after these should the relationships among the three variables be explored. However, the manuscript focuses solely on the latter, with a disproportionate emphasis on the results of the Confirmatory Factor Analysis (CFA), which is less appropriate in this section.
- Tables:
- Table 1: Remove the r² values.
- Table 2: The formatting is incorrect.
- There is redundancy in data between Table 1 and Table 2—consider merging the tables.
- There are discrepancies in correlation values (e.g., PECPB and SS are reported as 0.01* in Table 1 but -0.099** in Table 2). Please verify and correct.
- For CFA results, a table summarizing model fit indices would be clearer. The results could be briefly described and reported under the Methods section instead.
- Discussion
- Some interpretations are problematic. For example, in lines 331–335, the finding that bullying experiences are associated with school burnout and satisfaction should not be equated with other studies concluding that peer bullying threatens students’ psychological and social health, as these refer to different constructs.
- The claim that PE classes have different consequences compared to other subjects is unsupported, as no comparative data from other courses are presented.
References
The literature should be updated, particularly in the following sections: line 86, line 344, and line 364. Please ensure that more recent and relevant references are incorporated to strengthen the study's theoretical foundation and discussion.
Comments on the Quality of English LanguageThe manuscript is generally well-written, with clear and appropriate use of academic English. Only minor language edits are needed.
Author Response
April 3, 2025
Dear Reviewer
Manuscript ID:
healthcare-3540329
Manuscript title:
Perceived bullying in physical education classes, school burnout and satisfaction: a contribution to understand – Children school well-being
My colleagues and I would like to thank you for the opportunity to resubmit our manuscript to Healthcare. We found that the academic reviewers’ comments were very helpful, and we have done our best to incorporate all their suggestions and reply to the reviewers’ comments. We believe that this has made a significant contribution to the overall quality of the manuscript.
Thank you for considering our manuscript.
On behalf of all authors, yours sincerely,
Comments from Reviewer 2
Thank you for the opportunity to review the topic of the manuscript, “Perceived Bullying in Physical Education Classes, School Burnout and Satisfaction: A Contribution to Understand –Children School Well-Being.”, addresses an important issue regarding students' psychological experiences in school. The objective to examine the impact of peer bullying in physical education (PE) classes on school burnout and satisfaction among middle students is valuable. However, there are several concerns related to the structure, clarity, methodology, and interpretation of results that need to be addressed to enhance the rigor and focus of the manuscript.
Authors' response:
- Manuscript Structure
The manuscript should follow the conventional structure: Introduction, Methods, Results, Discussion, and Conclusion.
Authors' response:
We thank the reviewer for this helpful structural suggestion. In response, we have revised the manuscript to align more closely with the conventional scientific format. Specifically, we reorganized the text into clearly defined sections following the standard structure: Introduction, Methods, Results, Discussion, and Conclusion.
Furthermore, we ensured that the manuscript now adheres to the structural and formatting guidelines of the journal. We believe that these revisions have improved the clarity, coherence, and overall academic rigor of the paper.
Sections currently labeled as 3. Procedures, 4. Instruments, and 5. Statistical Analyses should be consolidated under section 2. Methods.
Authors' response:
Thank you for this precise and helpful recommendation. In accordance with the reviewer’s suggestion, we have consolidated the previously separate sections titled 3. Procedures, 4. Instruments, and 5. Statistical Analyses under a single, unified section titled 2. Methods. Each of these components is now presented as a clearly labeled subsection within the Methods section , in line with the journal's formatting guidelines. We believe this revision improves the structural coherence and readability of the manuscript.
- Introduction
Line 49: The citation format is incorrect. It should be written as [1].
Authors' response:
We adapted the references to the format of the journal.
The impact of peer bullying is repeated in the second and third paragraphs—please revise to avoid redundancy.
Authors' response:
We thank the reviewer for pointing out the redundancy between the second and third paragraphs regarding the impact of peer bullying. In response, we carefully revised these sections by eliminating overlapping statements and consolidating the ideas into a more concise and integrated narrative. The revised version now presents a more coherent explanation of the psychological and academic consequences of peer bullying, while maintaining the key arguments and supporting references
“Students exposed to peer bullying often experience decreased self-esteem, social with-drawal, and psychological problems such as anxiety, depression, and even post-traumatic stress [7–15]. These effects can persist into later stages of education and significantly undermine victims’ academic performance, social adjustment, and over all well being. While sport-based programs have been proposed as a potential preventive strategy emphasizing awareness, inclusion, and social skill development [16,17]—physical education and sport (PES) classes can paradoxically become a setting where bullying is more frequent than in other school contexts. Competitive dynamics, insufficient supervision, and skill disparities often create conditions that foster bullying behaviors [18–20]. These experiences can negatively impact not only students’ physical and mental health, but also their school satisfaction, engagement, and overall learning experience [21,22].”
The statement “The literature suggests that sport-based practices may have a potential impact on preventing peer bullying” lacks a supporting citation—please provide references.
Authors' response:
Thank you for your observation. Upon review, we acknowledge that the statement regarding sport-based practices lacked sufficient empirical support within the current scope of our study. To maintain the clarity and focus of the manuscript, we have decided to remove this sentence from the revised version. We appreciate the reviewer’s attention to accuracy and scientific rigor.
The claim that “Peer bullying in PES courses has a different structure than bullying in other courses” is unclear, especially since the forms of bullying mentioned (physical, verbal, psychological) are similar. Please clarify and support this claim with literature.
Authors' response:
Thank you for your comment. We agree that the distinction between peer bullying in PES courses and in other subject areas requires clear articulation.
As noted in the manuscript, we addressed this point by highlighting contextual factors specific to PES—such as physical performance, body image, and the visibility of individual skills—that increase social comparison and competition among students. These conditions may foster unique bullying dynamics, including ridicule and exclusion targeting students with lower physical abilities [23–25].
“This context paves the way for bullying incidents to occur more frequently and directly. Students who exhibit low skills, especially in activities requiring physical performance, are more exposed to behaviors such as ridicule, exclusion, or humiliation by other students [23,24]. For example, Bejerot et al. [25] reported that 48.6% of adults with below-average motor skills were bullied at school and pointed out that low physical abilities increase the risk of victimization.”
Key terms such as School Burnout and School Satisfaction are not defined. A thorough literature review explaining the empirical relationships among the study variables is also missing.
Authors' response:
Thank you for this important observation. In response, we have revised the introduction to include clear definitions of the key constructs—school burnout and school satisfaction. We have also expanded the literature review to better explain the empirical relationships between peer bullying, burnout, and satisfaction.
“School burnout refers to students' feelings of emotional exhaustion, cynicism toward academic tasks, and a reduced sense of accomplishment in school settings [30]. School satisfaction, on the other hand, is defined as students' overall cognitive and emotional appraisal of their school experience, including their sense of belonging and perceived support [31]. Previous research has consistently shown that exposure to peer bullying is associated with increased levels of school burnout and reduced school satisfaction among students. Studies have found that victims of bullying often report higher emotional exhaustion, lower motivation, and weaker feelings of belonging at school [32].
It is unclear whether the School Burnout Scale, which includes subscales such as “emotional exhaustion,” “depersonalization,” and “low expectation of success,” accurately captures the construct of school burnout in this study. Please justify the choice of scale.
Authors' response:
Thank you for this insightful comment. The School Burnout Scale used in the present study was selected based on its strong theoretical foundation and widespread use in the school burnout literature. The scale includes the subdimensions of emotional exhaustion, depersonalization, and low expectation of success, which together provide a comprehensive assessment of students’ emotional and cognitive disengagement from school tasks. These dimensions are conceptually aligned with the multidimensional construct of school burnout as originally proposed by Salmela-Aro et al. [30], and have been validated in various cultural contexts, including in Turkey. We have now elaborated on this rationale in the manuscript and provided the appropriate citations to support the use of this measurement tool.
We used the scale in a holistic structure as unidimensional rather than 3-dimensional. In addition, we tested its unidimensional structure with CFA analysis in this study.
- Methods
Participants: While participant information is described for the first stage, the second stage includes the primary sample. Demographic characteristics should be moved to the Results section.
Authors' response:
Thank you for this observation. While we understand the common practice of reporting demographic characteristics in the Results section, we believe that in the context of this study, it is more appropriate to present them in the Participants subsection. This allows readers to immediately understand the sample composition alongside the data collection stages, especially given the two-phase structure of the study.
Keeping the demographic information here helps maintain transparency about the characteristics of both the preliminary and main samples, which we believe enhances the methodological clarity of the manuscript.
However, we have ensured that this section remains concise and clearly structured to avoid redundancy later in the Results section.
Lines 172–173: These lines contain duplicate content and should be revised.
Authors' response:
Thank you for pointing this out. The repeated sentence in lines 172–173 was included unintentionally. We have now removed the duplicate content from the revised manuscript. We appreciate the reviewer’s attention to detail, which helped improve the clarity and precision of the text.
Instruments:
PECPB Scale: The development process is described, but the actual items and the types of bullying measured are not presented. It is unclear whether the scale includes dimensions such as physical, verbal, psychological, or cyberbullying. The scoring method and interpretation of scores must also be clarified.
Authors' response:
Thank you for your helpful comment regarding the PECPB Scale. In the revised manuscript, we clarified that the scale is designed as a brief, unidimensional tool to measure perceived peer bullying specifically within the physical education setting. Given the target population (middle school students), a short and focused scale was intentionally developed to ensure cognitive accessibility and minimize response fatigue. The unidimensional structure is supported by the results of our exploratory and confirmatory factor analyses, which indicated that the items load onto a single factor. The scale addresses common manifestations of bullying in PE classes—such as verbal mockery, social exclusion, and physical intimidation—within a single construct of perceived bullying.
“Higher average scores indicate a greater level of perceived peer bullying in physical education classes. An example item is: “My classmates make fun of my performance during physical education courses”
School Burnout Scale and School Satisfaction Scale: Please report the Cronbach’s alpha values obtained in this study. Provide one sample item from each subscale. Scoring methods and score interpretations should be clearly explained.
Authors' response:
Thank you for your valuable comment regarding the School Burnout Scale and the School Satisfaction Scale. In response, we have revised the manuscript to provide additional methodological details as requested.
For the School Burnout Scale, we clarified that the scale was used as a unidimensional construct in the present study. Accordingly, a second-order confirmatory factor analysis (CFA) was conducted, and the results supported the unidimensional structure with acceptable fit indices. We also reported the Cronbach’s alpha value (α = .899), indicating high internal consistency. Additionally, we included one sample item from the scale: “I feel overwhelmed by schoolwork.”
The scoring method (5-point Likert scale ranging from 1 = Strongly disagree to 5 = Strongly agree) and the interpretation (higher scores indicate higher levels of burnout) have been clearly stated in the revised version.
We appreciate the reviewer’s feedback, which helped us strengthen the clarity and completeness of the methods section.
Line 226: Correct the typo from “theSopper´s online” to “the Sopper´s online.”
Authors' response:
Thank you for your attention. Necessary correction were made.
The content under 5.1 Preliminary Analysis should be moved to the Participants section as it explains sample size estimation.
Authors' response:
Added to Participants section.
- Results
Lines 260–271: This section should focus solely on statistical results. Descriptions of statistical methods belong in the Statistical Analysis section.
Authors' response:
Thank you for your helpful observation. In accordance with the reviewer’s recommendation, we have revised lines 260–271 by removing the explanatory content related to statistical procedures from the Results section. These descriptions have been appropriately relocated to the Statistical Analysis subsection within the Methods section.
As a result, the Results section now focuses solely on reporting the statistical outcomes of the analyses conducted. We believe this revision has improved the clarity, structure, and readability of the manuscript. We sincerely appreciate the reviewer’s contribution to enhancing the quality of our work.
The results should be presented in accordance with the research objective, beginning with the participants’ demographic characteristics, followed by the status of peer bullying, school burnout, and school satisfaction. Only after these should the relationships among the three variables be explored. However, the manuscript focuses solely on the latter, with a disproportionate emphasis on the results of the Confirmatory Factor Analysis (CFA), which is less appropriate in this section.
Authors' response:
Thank you for this thoughtful observation. We agree that Confirmatory Factor Analysis (CFA) related to measurement tools is typically more appropriate within the Methods section. Accordingly, the CFA results related to the validation of the measurement instruments (School Burnout Scale, School Satisfaction Scale, and the PECPB Scale) have been presented under the Methods section.
However, the CFA presented in the Results section serves a different purpose: it was conducted as part of the Structural Equation Modeling (SEM) process to validate the overall structure of the proposed model and test the relationships among latent variables. Since this CFA pertains directly to the model tested in the main analysis, we believe it is appropriate to present it in the Results section as a preliminary step before reporting the path coefficients.
We have revised the section to clarify this distinction and appreciate the reviewer’s attention, which helped us improve the organization and clarity of the manuscript.
Tables:
Table 1: Remove the r² values.
Authors' response:
Thank you for your careful review. We noticed that an additional table had been mistakenly included in the previous version of the manuscript. This table has now been removed in the revised version. We appreciate the reviewer’s attention to detail, which helped improve the overall quality and clarity of the manuscript.
Table 2: The formatting is incorrect.
Authors' response:
Thank you for your comment. The formatting issues in Table 2 have been corrected in the revised version of the manuscript. We appreciate your feedback and attention to detail.
There is redundancy in data between Table 1 and Table 2—consider merging the tables.
Authors' response:
Thank you for your observation. We noticed that an incorrect version of a table had been mistakenly included in the previous submission. In the revised manuscript, the incorrect table has been removed, and only the correct version is retained. We appreciate the reviewer’s careful attention.
There are discrepancies in correlation values (e.g., PECPB and SS are reported as 0.01* in Table 1 but -0.099** in Table 2). Please verify and correct.
Authors' response:
Thank you for your careful observation. The discrepancy in the correlation values between PECPB and School Satisfaction has been corrected in the revised version of the manuscript. The updated tables now reflect the accurate results from the final analysis. We appreciate your attention to detail.
For CFA results, a table summarizing model fit indices would be clearer. The results could be briefly described and reported under the Methods section instead.
Authors' response:
Thank you for your helpful suggestion. We agree that presenting model fit indices in a clear and organized way is important for readability. While we considered formatting the CFA results as a table, we ultimately decided to retain them in a concise paragraph form, as the number of indices reported is limited and the values are straightforward.
We have ensured that the results are now presented in a clear and easily readable manner within the Methods section, in line with the purpose of confirming the validity of the measurement tools. We hope this presentation still meets the clarity expected and sincerely appreciate the reviewer’s thoughtful feedback.
- Discussion
Some interpretations are problematic. For example, in lines 331–335, the finding that bullying experiences are associated with school burnout and satisfaction should not be equated with other studies concluding that peer bullying threatens students’ psychological and social health, as these refer to different constructs.
Authors' response:
Thank you for this valuable comment. We acknowledge the distinction between the constructs examined in our study namely, school burnout and school satisfaction and the broader concepts of psychological and social health referenced in other studies.
In response, we have revised the paragraph to avoid conflating these constructs. The updated version now clearly states that while our findings are thematically aligned with the literature highlighting the negative impact of peer bullying on student well-being [30,58-60], the outcomes measured in our study are specifically related to educational contexts. We believe this revision provides a more accurate and conceptually precise interpretation of the findings, and we sincerely thank the reviewer for their insightful observation.
“The findings of the study show that bullying experiences in PES courses weaken students' commitment to school and increase their school burnout levels. While these educational outcomes are distinct from broader measures of psychological and social health, the findings are in line with existing literature emphasizing the negative impact of peer bullying on various aspects of student well-being [30,58-60]. This study contributes to the literature by offering a course-specific perspective, suggesting that bullying may have unique manifestations and consequences in physical education and sport (PES) classes due to their competitive, performance-based, and socially visible nature. In such settings, where physical competence is prominent and the need for social approval is heightened, bullying appears to particularly affect students’ experiences of school satisfaction and burnout.”
The claim that PE classes have different consequences compared to other subjects is unsupported, as no comparative data from other courses are presented.
Authors' response:
Thank you for this insightful observation. We agree that the study does not include comparative data from other subject areas, and thus, definitive claims about the distinctiveness of PE classes relative to other courses should be avoided. In response, we have revised the relevant statements in the discussion to clarify that while the findings suggest bullying may take unique forms in the context of PES, this study does not provide comparative evidence. We have added a sentence acknowledging this limitation and recommending future research to explore differences across subject areas. We appreciate the reviewer’s comment, which helped us refine the accuracy and scope of our interpretations.
“However, it is important to note that this study did not include data from other subject areas; therefore, conclusions about the distinctiveness of PE-related bullying outcomes should be interpreted cautiously. Future research could benefit from cross-subject comparisons to better understand whether the dynamics and effects of bullying in PE classes are indeed distinct from those observed in other educational settings.”
References
The literature should be updated, particularly in the following sections: line 86, line 344, and line 364. Please ensure that more recent and relevant references are incorporated to strengthen the study's theoretical foundation and discussion.
Authors' response:
Thank you for your suggestion to include more recent and relevant literature.
Corrections for 86 line are below
“Recent findings indicate that poor fine motor skills at age 3 were significantly associated with peer victimization at age 5. Moreover, while both fine and gross motor difficulties at age 5 showed concurrent associations with peer victimization, gross motor impairments had a stronger impact. Notably, only fine motor difficulties at age 5 were significantly linked to peer victimization reported at age 8 [26]. A recent study by Bejerot, Ståtenhag, and Glans [25] found that adults diagnosed with ADHD who recalled having below-average motor skills reflected in poor performance in PE classes were significantly more likely to report being victims of bullying during childhood. This association was especially strong during early school years and continued through adolescence. However, no significant relationship was found between poor motor skills and perpetrating bullying. These findings underscore the role of physical competence, particularly in PE settings, as a risk factor for peer victimization”
Corrections for 344 line are below
“Recent research has also highlighted that peer victimization in school settings can reduce students' life satisfaction and academic self-efficacy. For instance, a large-scale study among middle school students in Switzerland showed that supportive school environments and a strong sense of peer belonging were associated with reduced victimization and higher life satisfaction [63]. Chen et al. [64] conducted a large-scale, cross-national study using PISA 2015 and 2018 data from five countries (China, Japan, South Korea, the US, and the UK) to investigate how peer victimization affects adolescents' school belonging, truancy, and life satisfaction. The findings revealed that peer victimization had a significant negative direct effect on school belonging, which in turn mediated both increased truancy and decreased life satisfaction. Huang [65] found that peer victimization was negatively associated with students’ academic life satisfaction and school belonging, and positively associated with schoolwork-related anxiety. These findings are consistent with the current study’s results, highlighting how peer bullying in school contexts can reduce students’ emotional engagement and overall satisfaction with school.”
Corrections for 364 line are below
“For example, earlier research by Puhl and Luedicke [73] indicated that students who experienced bullying related to their physical abilities tended to avoid physical activities and disengage from general school participation. Supporting this, Benítez-Sillero et al. [74] found that lower levels of cardiorespiratory fitness were significantly associated with victimization, whereas higher muscular strength—particularly in boys—was positively related to bullying perpetration.”
Round 2
Reviewer 1 Report
Comments and Suggestions for Authors
I believe the authors’ review work has been very thorough. I found their responses particularly satisfying, and I think the paper has significantly improved, especially in its theoretical aspects and limitations. It can be accepted as it is.
Author Response
Dear reviewer,
We are sincerely grateful for your recommendation to accept the manuscript for publication. Your thoughtful evaluation and encouraging feedback are deeply appreciated.
Best regards,
Reviewer 2 Report
Comments and Suggestions for Authors
Dear Author,
Thank you for the opportunity to review your revised manuscript titled "Perceived Bullying in Physical Education Classes, School Burnout and Satisfaction: A Contribution to Understand Children’s School Well-Being."
This is the second time I have read your article. I appreciate the efforts made in the revision, and I acknowledge that the overall quality has improved. However, there remain some critical gaps that limit the manuscript’s contribution to the understanding of children’s school well-being, particularly in the context of physical education (PE) classes.
The manuscript emphasizes that students in PE settings may experience unique bullying behaviors due to factors such as physical performance, body image, and personal skills. This is an important and timely topic. However, the current version of the manuscript does not provide sufficient descriptive statistical analysis of participants’ demographic or background characteristics, such as physical fitness, self-perceived body image, or motor skills. Moreover, the types and prevalence of perceived bullying behaviors are not clearly delineated.
While the study highlights the relationship among perceived bullying, school burnout, and school satisfaction, it does not explore the specific nature of bullying behaviors nor how these distinct types influence psychological outcomes. As such, the findings fall short of substantiating the paper’s stated aim of contributing to a deeper understanding of children’s school well-being.
Additionally, the manuscript devotes a substantial portion to the development and validation of measurement instruments, which tends to overshadow the substantive examination of the research questions. The effort in instrument development is commendable. However, to align more closely with the stated aim of understanding children’s school well-being, it would be beneficial to place greater emphasis on interpreting the empirical relationships and conceptual implications of the findings.
To enhance the manuscript’s value and relevance, I suggest the authors:
- Include a more detailed description and analysis of participants’ demographic and background variables.
- Clearly identify and categorize the forms of perceived bullying relevant to the PE context.
- Elaborate on how these specific bullying experiences relate to school burnout and satisfaction.
- Rebalance the manuscript to ensure that the instrument development does not eclipse the discussion on children’s school well-being.
Thank you again for your submission. I encourage further refinement so that the paper may better fulfill its potential contribution to this important area of educational and psychological research.
Author Response
Dear reviewer,
We would like to express our sincere gratitude for your insightful comments and constructive suggestions, which have significantly enhanced the quality of the present manuscript. Below, We provide detailed responses to each of your observations.
Comments from Reviewer 2
The manuscript emphasizes that students in PE settings may experience unique bullying behaviors due to factors such as physical performance, body image, and personal skills. This is an important and timely topic. However, the current version of the manuscript does not provide sufficient descriptive statistical analysis of participants’ demographic or background characteristics, such as physical fitness, self-perceived body image, or motor skills. Moreover, the types and prevalence of perceived bullying behaviors are not clearly delineated.
Authors' response:
Thank you for this valuable and constructive feedback. In response, we have made the following revisions and clarifications within the manuscript:
Clarification on Sample Background Characteristics:
A sentence was added to the Participants section to provide more context about the nature of the sample and its demographic diversity. Specifically, we noted that: “Participants were selected from public schools with average socio-demographic profiles located in five different provinces across Türkiye, each representing a different geographical region.”
Addressing the Omission of Variables Such as Physical Fitness or Body Image:
We acknowledge that the study did not collect direct data on physical fitness, body image perception, or motor skills. In light of this, we included an explicit statement in the Limitations section to clarify this point:
“In addition, the study did not include background variables such as students’ physical fitness, body image perception, or motor skill levels. These characteristics may be associated with bullying experiences in PE settings, and their exclusion limits the depth of interpretation regarding individual differences. Future research is encouraged to integrate such variables to provide a more holistic understanding of perceived bullying and student well-being in physical education contexts.”
Comments from Reviewer 2
The reviewer recommended that the authors clearly define and categorize the types of perceived bullying behaviors specific to physical education (PE) classes, in order to enhance the conceptual clarity and contribution of the study.
Authors' response:
We fully agree that understanding specific types of bullying could enrich the interpretation of the findings. However, based on both theoretical and empirical grounds, we opted for a unidimensional structure in the development of the Perceived Bullying in Physical Education Scale.
Specifically:
During Exploratory Factor Analysis (EFA) and Confirmatory Factor Analysis (CFA), the items loaded strongly on a single factor, and attempts to extract multiple factors did not yield a psychometrically sound structure.
Additionally, the internal consistency reliability and fit indices of the one-factor model were within acceptable to good ranges, supporting the use of a unidimensional scale.
From a conceptual standpoint, perceived bullying was operationalized as a global construct encompassing students’ subjective experiences of peer victimization during PE classes, rather than as predefined behavioral subtypes.
Furthermore, the literature on peer bullying includes several validated unidimensional measures, especially when the goal is to capture perceived victimization rather than classify behavior types.
In light of this, while we acknowledge the potential value of multi-categorical classification in future research, we believe that maintaining a unidimensional scale is more appropriate and methodologically justified for the current study’s aims and empirical structure.
Comments from Reviewer 2
Elaborate on how these specific bullying experiences relate to school burnout and satisfaction.
Authors' response:
Thank you for this valuable comment. We have expanded the discussion section to more clearly articulate how perceived bullying in Physical Education (PE) classes impacts school burnout and satisfaction. Specifically, we added a summarizing sentence to the first paragraph of the discussion section, emphasizing the psychological mechanisms—such as decreased sense of belonging, self-worth, and perceived competence—through which bullying in PE settings contributes to increased school burnout and reduced school satisfaction.
Additionally, we supported this explanation by including new references that highlight the mediating roles of psychological vulnerability and motivational decline in bullying contexts (Hoferichter et al., 2021; Vasconcellos et al., 2020; Zhao et al., 2024).
Comments from Reviewer 2
"Rebalance the manuscript to ensure that the instrument development does not eclipse the discussion on children’s school well-being."
Authors' response:
Thank you for your valuable evaluation and suggestions. In the Methods and Results sections, only the essential information regarding the scale development process has been included, and the level of detail has been deliberately kept to a minimum.
Additionally, in line with your previous recommendations, the Introduction and Discussion sections have been restructured to better reflect the research focus. The effects of perceived bullying in physical education and sports (PES) classes on students' school burnout and school satisfaction have been discussed in comparison with both national and international literature, and the practical implications of the findings have been emphasized. In the Discussion section, the interpretation of the findings and their conceptual implications were prioritized over the technical details of the scale development process.
With these revisions, while the scale development remains an essential component of the study, it no longer occupies a central position in the overall structure of the manuscript, ensuring a more balanced presentation.
Round 3
Reviewer 2 Report
Comments and Suggestions for Authors
Dear Authors,
Thank you for your continued efforts in revising the manuscript entitled “Perceived Bullying in Physical Education Classes, School Burnout and Satisfaction: A Contribution to Understand Children’s School Well-Being.” This is my third time reviewing your manuscript, and I commend you for the notable improvements made thus far.
Overall, the writing is clearer and more coherent; however, several grammatical and syntactical issues remain throughout the text. For instance, in the abstract, the phrase “his study examines...” should be corrected to “This study examines...”. Additionally, expressions such as “the bullying perceived by students” can be made more concise and idiomatic by using “students’ perceived bullying.”
I recommend that the manuscript undergo a comprehensive English-language proofreading to enhance clarity, fluency, and overall professionalism.
Author Response
Dear reviewer,
We would like to express our sincere gratitude for your insightful comments and constructive suggestions, which have significantly enhanced the quality of the present manuscript. As requested, the article has undergone a comprehensive English language review by an accredited entity, which provided a revision certificate that is attached.
Thank you for considering our manuscript.
On behalf of all authors, yours sincerely,
Pedro Duarte-Mendes